# Induction of *PLXNA4* Gene during Neural Differentiation in Human Umbilical-Cord-Derived Mesenchymal Stem Cells by Low-Intensity Sub-Sonic Vibration

**DOI:** 10.3390/ijms23031522

**Published:** 2022-01-28

**Authors:** Hyunjin Cho, Hee-Jung Park, Young-Kwon Seo

**Affiliations:** 1Research Institute of Integrative Life Sciences, Dongguk University, Goyang-si 10326, Korea; dr.hjcho@hanmail.net; 2Department of Medical Biotechnology (BK21 Plus Team), Dongguk University, Goyang-si 10326, Korea; gnflwldk98@naver.com

**Keywords:** hUC-MSCs, low-intensity sub-sonic vibration, *PLXNA4*, *SEMA3A*, neural signaling

## Abstract

Human umbilical-cord-derived mesenchymal stem cells (hUC-MSC) are a type of mesenchymal stem cells and are more primitive than other MSCs. In this study, we identify novel genes and signal-activating proteins involved in the neural differentiation of hUC-MSCs induced by Low-Intensity Sub-Sonic Vibration (LISSV). RNA sequencing was used to find genes involved in the differentiation process by LISSV. The changes in hUC-MSCs caused by LISSV were confirmed by *PLXNA4* overexpression and gene knockdown through small interfering RNA experiments. The six genes were increased among genes related to neurons and the nervous system. One of them, the *PLXNA4* gene, is known to play a role as a guide for axons in the development of the nervous system. When the PLXNA4 recombinant protein was added, neuron-related genes were increased. In the *PLXNA4* gene knockdown experiment, the expression of neuron-related genes was not changed by LISSV exposure. The *PLXNA4* gene is activated by sema family ligands. The expression of *SEMA3A* was increased by LISSV, and its downstream signaling molecule, FYN, was also activated. We suggest that the *PLXNA4* gene plays an important role in hUC-MSC neuronal differentiation through exposure to LISSV. The differentiation process depends on *SEMA3A*-*PLXNA4*-dependent FYN activation in hUC-MSCs.

## 1. Introduction

The conversion of mechanical stimuli into biochemical information within cells provides biological activation or inactivation signals in various cell types. Mechanical stimulation is a kind of physical force and has various effects on cells through the regulation of cell proliferation and differentiation. Among categories of cells, mesenchymal stem cells (MSCs) are differentiated into various types and have great potential for tissue engineering.

Although many studies on mechanical stimuli have been reported, few have examined the effect on MSCs. In vascular tissue engineering studies, bone marrow-derived MSCs (BM-MSCs) were differentiated into endothelial-like cells by shear stress [1,2,3], and these results were similar to other tissue-derived MSCs, such as human placenta and adipose-derived MSCs [4]. In cartilage tissue engineering, the combined treatment of MSCs with chondrocytes improves symptoms of knee cartilage defects [5], and Lin et al. reported a synergistic effect of cartilage regeneration by dynamic compression in BM-MSCs [6]. Most of the studies were performed by adding enzymes, cytokines, and growth factors that support the potential regenerative capacity of MSCs, in addition to mechanical stimulation [4,5,6,7].

Sub-sonic refers to a frequency that is so low that it is inaudible and slower than the speed of sound. Low-intensity vibration (LIV) is a stimulus that transmits vibration with intensity in the range of 20 to 200 Hz frequencies. LIV can be effective in improving the bone and muscle index at the tissue level [8,9,10] and reducing myeloma cell-induced osteoclast formation [11]. Furthermore, LIV decreased the pro-inflammatory cytokines IL-6, IFN-γ, and TNF-α in cultured murine macrophages [12] and inhibited tumor progression [13].

Plexin-A4 (encoded by *PLXNA4*) belongs to the plexin A family. It binds neuropilin to propagate signals with class 3 semaphorins (*SEMA3*) into neurons and plays an important role in the induction of axons to their synaptic target during neural development [14,15,16,17]. This complex activates the cdk-5-mediated isoform A of phosphatidylinositol 3-kinase enhancer and promotes glioma cell invasion through the modulation of Akt activity [18,19]. In a recent report, Plxna4 variants were found to be involved in Alzheimer’s disease pathogenesis through amyloid beta deposition [20].

Several studies have examined the effects of mechanical stimulation alone on hMSCs and other cells. Previous work by our group and others reported the effects of mechanical stimulation in hMSCs [21,22,23,24]. Neural differentiation of hMSCs was induced by sonic vibration or/and electromagnetic fields, allowing therapeutic applications in models of spinal cord injury and ischemic stroke [25,26]. This mechanical stimulation equally induced neural differentiation of hMSCs, and in particular, morphological changes similar to the neural cell shape were dramatically induced by sub-sonic vibrations [21]. In the present study, we aimed to investigate key genes involved in this differentiation by low-intensity sub-sonic vibration (LISSV) through RNA sequencing based on next-generation sequencing.

## 2. Results

### 2.1. RNA Sequencing Confirmed That Six Genes Were Increased by LISSV

In a previous study, we reported the effect of LISSV on hUC-MSCs [21]. A morphological change in the cells was induced and the shape was close to that of neurons. To identify novel genes involved in this change, RNA-sequencing analysis was performed. In the gene expression analysis related to the cell differentiation of hUC-MSCs, 16 genes increased more than threefold, and among them, 5 genes increased during LISSV, including *PLXNA4*, *FMN1*, *AREG*, *STMN2*, and *SERPINI1* (Figure 1). Protein levels were detected in a time-dependent manner. The levels of *PLXNA4* increased, but *FMN1* decreased, and levels of *SERPINI1* were unchanged (Figure 2).

### 2.2. Recombinant PLXNA4 Protein Affected Neural Differentiation of hUC-MSCs

Plexins are proteins involved in axon growth and are expressed on the surface of axon growth cones. Nine genes have been identified. *PLXNA4* belongs to class A along with *PLXNA1, PLXNA2*, and *PLXNA3*. Plexin activation in growth cones causes actin and microtubule destabilization and endocytosis, involved in the contraction of growth cone protrusions [27]. The recombinant PLXNA4 protein was used for PLXNA4 overexpression analysis. The recombinant PLXNA4 protein was non-toxic in hUC-MSCs up to 2 μg/mL and significantly reduced the number of viable cells to 77% at 3 μg/mL (data were not shown). To demonstrate the effect on hUC-MSCs, the morphological changes in hUC-MSCs were observed daily after the addition of the recombinant PLXNA4 protein. Figure 3a shows that the amount of *PLXNA4* gene expression increased after treatment with recombinant PLXNA4. On the fourth day, the inhibition of cell proliferation was confirmed by MTT assay (Figure 3b), and a morphological change was observed on the fifth day (Figure 4). The cells did not proliferate when treated with the recombinant PLXNA4 protein compared to controls, as they did with LISSV exposure. The cell body was brighter than that of the control group, and the filamentous shape was similar to that of the neuron. The expression of neuron-related genes by the PLXNA4 protein addition was confirmed, and the expression of most neuron-related genes, such as *MAP2, NEUROD1,* and *NF-L,* was increased, but for *MBP* it was unclear (Figure 5). Voltage-gated calcium (Ca^2+^) channels are key channels that induce changes in the membrane potential, and intracellular Ca^2+^ transients signal the initiation of many physiological events. The Cav2 subfamily is primarily responsible for the initiation of synaptic transmission [28]. The recombinant PLXNA4 protein has induced the expression of Cav2.1 and Cav2.2 genes in hUC-MSCs (Figure 6).

### 2.3. The Neural Differentiation of hUC-MSCs by LISSV Was Not Induced upon PLXNA4 Gene Silencing

siRNA is a gene-silencing assay in which the RNA-induced silencing complex binds to the target mRNA and interferes with the synthesis of the target protein. To evaluate the role of *PLXNA4* in neuronal differentiation, we performed a silencing *PLXNA4* expression experiment. We tested the cytotoxicity of siRNA duplex transfection in hUC-MSCs before siRNA experiments. When treated with 90 nmol of siPLXNA4 duplex transfection, viable cells were 93.1% after siNegative control transfection and 90.4% after siPLXNA4 transfection (Figure 7). Figure 8 shows a morphological change or *PLXNA4* level change in the case of silencing *Negative* (siNegative) control and *PLXNA4* (siPLXNA4) expressions, which occurred with or without LISSV exposure. A universal, scrambled, negative control siRNA duplex was used as the silencing *Negative* (siNegative) control. This duplex and the transfection regents did not affect cells when exposed to LISSV compared to controls. The silencing *Negative* (siNegative) group with LISSV expressed the same level of *PLXNA4* as the only LISSV-treated group, whereas the silencing *Negative* (siNegative) and *PLXNA4* (siPLXNA4) groups with (+) or without (−) LISSV expressed no *PLXNA4*. The *PLXNA4* gene was not expressed even in the silencing *PLXNA4* (siPLXNA4) expression group with LISSV. LISSV exposure increased *PLXNA4* levels in the silencing *Negative* (siNegative) control group but not in the silencing *PLXNA4* (siPLXNA4) group. LISSV affects *PLXNA4* gene expression, and *PLXNA4* gene expression is thought to be associated with neural differentiation in hUC-MSCs. The expression of neuron-associated genes was analyzed by RT-qPCR in a *PLXNA4* gene-silencing study. When the *PLXNA4* gene was knocked down in hUC-MSC, the expression of neuronal-related genes, such as *NF-L*, *MBP*, and *MAP2* genes, was not increased by LISSV (Figure 9).

### 2.4. Changes with LISSV versus PLXNA4 in Sema-Dependent Signaling

Our results showed that the *PLXNA4* gene is associated with neuronal differentiation of hUC-MSCs. In a previous study, we reported the differentiation of hUC-MSCs by LISSV [21]. The differentiation was induced in the LDMEM medium without any supplements for neural induction. The differentiation was non-specific, so genes related to oligodendrocytes, astrocytes, and neurons were all expressed. To compare neural differentiation patterns between LISSV and PLXNA4, we studied three types of neural-specific gene expression. As shown in Figure 10, the induction of neural differentiation by LISSV was non-specific, but only the *MAP2* and *NEUROD1* genes were expressed by the PLXNA4 recombinant protein. Differentiation of hUC-MSCs by the PLXNA4 recombinant protein is thought to be induced specifically in neurons.

We thought that MSC-specific markers were no longer expressed when differentiation was induced by both LISSV and recombinant PLXNA4 protein. CD73, CD105, and CD90 are markers specifically expressed in human MSCs [29]. In Fluorescence-activated cell sorting (FACS) analysis, we analyzed anti-CD73 and anti-CD105 expression after LISSV and recombinant PLXNA4 protein treatment (Figure 11). Before treatment, anti-CD73 was expressed at 97.8% in hUC-MSCs and anti-CD105 was expressed at 81.3%. Before treatment, anti-CD73 was expressed at 97.8% and anti-CD105 was expressed at 81.3% in hUC-MSCs. When LISSV was given for 3 days, the expression of anti-CD73 was reduced to 90.4%, and anti-CD105 was reduced to 62.5%. In the case of PLXNA4 treatment, the expression of anti-CD73 was reduced to 83.2%, like LISSV, and anti-CD105 was also reduced to 62.5%. Both LISSV and recombinant PLXNA4 protein treatment reduced the expression of MSC-specific markers, suggesting that the differentiation process of hUC-MSCs is in progress.

Plexins are proteins involved in the signaling of the semaphorin family, and plxna4 belongs to class A of four types. Class A plexins interact with neuropilin co-receptor proteins, and specifically, the plxna4 and neuropilin signaling cascade can be activated by both SEMA3A and SEMA6A [30]. To identify semaphorins involved in inducing the expression of neuron-associated proteins in hUC-MSCs, we analyzed the expression of both *SEMA3A* and *SEMA6A*. *SEMA3A* was increased selectively (Figure 12) and its downstream signaling molecule, FYN, was activated in a time-dependent manner (Figure 13). Presynaptic vesicles for neurite outgrowth in neurons are induced through the activation of this signal. GAP43 regulates presynaptic vesicle interactions and SYN1 and synaptophysin proteins are presynaptic vesicle proteins. LISSV and the recombinant PLXNA4 protein increased the expression levels of *SYN1*, *GAP43*, and *synaptophysin* genes (Figure 14). The expression of neuron-related proteins was induced by activation of the semaphorin 3A-dependent plexin-A4 signaling cascade by LISSV in hUC-MSCs.

## 3. Discussion

In a previous study, we reported the neural differentiation of hUC-MSCs by LISSV. These changes are specifically due to mechanical stimulation in cells that were lacking growth factors and cytokines for the differentiation process. To find genes involved in this differentiation, RNA sequencing analysis was performed after LISSV exposure in hUC-MSCs.

RNA sequencing is a very useful technique for elucidating the presence and sequences of RNA in a sample using next-generation sequencing. This technique shows changes in the cellular transcriptome at a given moment. After 4 days of exposure to LISSV, the hUC-MSCs changed their shape to a neuron-like morphology, at which time the cells were harvested for RNA sequencing. First, we focused on upregulated genes, and among 16 genes, 5 genes were identified using real-time PCR: *PLXNA4, FMN1, AREG, STMN2*, and *SERPINI1*. Among these, four genes were expressed at the protein level in a time-dependent manner. *PLXNA4* increased gradually over 6 days, and *FMN1* decreased. The *PLXNA4* gene belongs to the plexin A family, and plexin A is a neuronal semaphorin receptor involved in axon guidance during neural development and neuron migration to synaptic organization [31,32,33,34]. In neurons, semaphorins transduce activation signals through plexin receptor proteins and the neuropilin family [35]. The recombinant PLXNA4 protein was used to study the role of the *PLXNA4* gene in hUC-MSC differentiation by LISSV. The *PLXNA4* gene was well expressed when 1.5 g or 2.0 g per well was added to hUC-MSCs. Cell proliferation was inhibited without dead cells. Thus, the reduction of cells by the PLXNA4 protein in the MTT assay indicates that hUC-MSCs have entered the differentiation process. After 5 days, a morphological change was induced and the morphology was close to that of a neuron. The cell body shone brightly, and two filaments extended in both directions around the cell body. Neuronal differentiation-related proteins such as MAP2, NEUROD1, and NF-L were expressed strongly in the immunofluorescence analysis, and the same results were confirmed in real-time PCR assays. These proteins and genes were also expressed in hUC-MSCs by LISSV [21], and the morphology of the differentiated cells was very similar in both cases. We observed that the differentiation process of hUC-MSCs induced by LISSV and *PLXNA4* was slightly different. In our previous report, when LISSV was given to hUC-MSCs, differentiation of hUC-MSCs was a neural non-specific process, so all three types of neural cell markers, astrocytes, oligodendrocytes, and neurons, were induced. However, PLXNA4-induced differentiation in hUC-MSCs is a neuron-specific process, as GFAP and MBP were not expressed. Voltage-gated calcium channels are the main mediators that allow calcium to flow into neurons when depolarization occurs. The Cav2.1 is a P/Q-type calcium channel, and Cav2.2 is an N-type calcium channel, while P/Q and N channels trigger neurotransmitter release [36]. The recombinant PLXNA4 protein induced neural differentiation while expressing Cav2.1 and Cav2.2.

hUC-MSCs are cells capable of self-renewal and differentiation into various lineages [37]. Wharton’s jelly derived from the human umbilical cord contains a higher amount of primitive MSCs compared to MSCs derived from bone marrow [38]. When MSCs begin to differentiate into cells of other lineages, those cells cannot maintain their stemness. Human MSCs express CD73, CD90, and CD105, but not CD34, CD45, or CD14 [39,40]. During both LISSV and recombinant PLXNA4 protein treatment, the expression of hUC-MSC specific markers, CD73 and CD105, was reduced. Therefore, those cells have begun to differentiate into other cells, particularly nerve-like cells.

To validate the function of the *PLXNA4* gene, *PLXNA4* gene silencing using siRNA analysis was performed. This assay was transient, and only 20–27 base pairs were used for gene silencing. After transfection of a specific gene base pair for an interfering RNA, the expression of the specific gene interferes with the complementary nucleotide sequence so the mRNA is degraded after transcription. In this assay, we used a transfection reagent for siRNA duplex transfection. The viable cells reached over 90% after transfection. We observed morphological changes and *PLXNA4* gene silencing in hUC-MSCs. After silencing, no change was noted in the level of *PLXNA4* gene expression in both the silencing *Negative* (siNegative) and *PLXNA4* (siPLXNA4) without LISSV. However, in the case of LISSV exposure, *PLXNA4* gene expression increased only in the silencing *Negative* (siNegative) expression group, and no change took place in the silencing *PLXNA4* (siPLXNA4) expression group. The same results were observed in the morphological change data, and in the case of silencing *PLXNA4* (siPLXNA4) expression, no morphological change occurred after exposure to LISSV. To confirm the neuronal differentiation after silencing *PLXNA4* (siPLXNA4) expression in hUC-MSCs, we analyzed the neuron-related gene expression by LISSV. After silencing *PLXNA4* (siPLXNA4) expression, the expression of neuron-related genes, *NF-L*, *MBP,* and *MAP2,* was unchanged compared to the controls and silencing *Negative* (siNegative) without LISSV groups. Consequently, we established that the *PLXNA4* gene is associated with the neural differentiation of hUC-MSCs by LISSV.

The interaction of the neuropilin (Nrp)/plexin receptor complex and semaphorins plays an important role in axonal development in the central nervous system. This complex is also involved in a variety of other developmental processes, spanning from cell polarization to migration to neuronal maturation [41]. The semaphorin protein family consists of eight classes and is found in vertebrates and invertebrates. Plexin receptors for SEMA are classified into four classes, and plexin-A4 interacts with specific SEMA classes to mediate signal activation. The Nrp/PLXNA4 receptor complex interacts with class 3 and class 6 semaphorins and is involved in axon guidance and anti-angiogenesis when interacting with class 3 semaphorins [42,43]. *SEMA3A*, one of the class 3 semaphorins, interacts with plexins-A1 and -A4 to induce cytoskeletal disruption, the inhibition of cell proliferation, and adhesion in endothelial cells [44]. *SEMA6A* acts as a chemical repellent for sympathetic axons and is involved in lamina-specific axon formation in the hippocampus [45,46]. To identify SEMAs interacting with PLXNA4 in hUC-MSCs by LISSV, we analyzed the gene expression of *SEMA3A* and *SEMA6A* after LISSV exposure. We found that *SEMA3A* is upregulated by 5-fold, but the level of *SEMA6A* is not changed. When the Nrp/plxna4 receptor complex interacted with *SEMA3A*, its downstream molecule, FYN, is recruited for signal activation, which is involved in the process of collapse for cell differentiation [47]. We confirmed that the FYN protein increases in a time-dependent manner. These results showed that increased levels of *PLXNA4* expression induced by LISSV in hUC-MSCs are associated with an increase in *SEMA3A*, and that neuronal differentiation of hUC-MSCs induced by LISSV is thought to be due to semaphorin 3A–plexin-A4-dependent signaling activation.

Sema3A regulates the density of the dendritic spine, small membrane protrusions from dendrites of neurons [48,49], and this signaling activation induces growth cone collapse and neuronal cells [41,50]. SYN1 and Synaptophysin are presynaptic vesicle proteins located in the cytoplasmic membrane of the presynapse, and GAP43, a neuromodulin, plays a role in regulating presynaptic vesicular function and axonal growth [51,52]. The expression of SYN1, GAP43, and synaptophysin was induced in hUC-MSCs through sema3 signaling activation upon treatment with LISSV and the recombinant PLXNA4 protein.

Many studies have reported the differentiation of MSCs, which is multi-day dependent and requires various growth and neurotropic factors and cytokines [53,54,55]. We reported neuronal differentiation of hUC-MSCs caused by LISSV alone, a mechanical stimulus. After 4–5 days, the shape of hUC-MSCs modified to neuron-like and neuron-related genes were expressed. This neuronal differentiation was activated through the Nrp/plxna4 receptor complex with the SEMA3A-dependent signaling mechanism. Discovering the specific mechanism that induces neural differentiation in hUC-MSCs via LISSV and applying it to neurodegenerative disorders will be very useful in stem cell therapy.

## 4. Materials and Methods

### 4.1. Cell Culture

hUC-MSCs were purchased from the American Type Culture Collection (ATCC, Washington, D.C., VA, USA) and were cultured in a nonhematopoietic (NH) stem cell medium (Miltenyi Biotech, Bergisch Gladbach, Germany) supplemented with 100 units/mL of penicillin and 100 μg/mL of streptomycin (Invitrogen, Carlsbad, CA, USA). Passage 6–10 cells were used in this experiment and the culture medium was replaced every 3–4 days. When hUC-MSCs reached approximately 80% density in 100 mm culture dishes, they were passaged in a 1:4 plate ratio using accutase (Innovative Cell Tech., San Diego, CA, USA) for cell isolation. One day before the experiment, the medium was replaced with low-glucose DMEM (LDMEM) supplemented with 10% FBS (Invitrogen, Carlsbad, CA, USA) followed by LISSV exposure.

### 4.2. LISSV Exposure

A Turbosonic Low-Intensity Sub-Sonic Vibrator (Turbosonic Korea, Seoul, Korea) was used [8]. The machine moves up and down creating undulating waves in the medium, which affects the cells. The hUC-MSCs were cultured at a low density and exposed to LISSV continuously for 4 days at a frequency of 30 Hz and acceleration of 13.5 to 14.1 after 1 day.

### 4.3. Cell Growth Assay

A cell growth assay was performed using 3-(4,5-dimethylthiazol-2-yl)-2,5-diphenyltetrazolium bromide (MTT) (Sigma-Aldrich, St. Louis, MO, USA) solution as an indicator of cell viability and proliferation. Viable cells containing NAD(P)H-dependent oxidoreductase enzymes were reduced to formazan by MTT. The MTT solution is added to cells in culture, at a final concentration of 0.83 mg/mL, and incubated for 3.5 h. At 570 nm, the absorbance was measured using a Versamax microplate reader (Molecular Device, Sunnyvale, CA, USA).

### 4.4. RNA-Sequencing Assay

The total RNA was extracted from the samples as above, and the concentration and purity were determined using a spectrophotometer with optical densities of A260 and A260/A280. RNA sequencing libraries of each sample were prepared using the TruSeq RNA Library Prep Kit (Illumina, San Diego, CA, USA), and the result was obtained using the Illumina HiSeq 2000 system. RNA-seq experiments were performed on hUC-MSCs exposed to LISSV for 4 days. The levels of gene expression were normalized to RPKM/FPKM (reads of paired-end fragments per kb of exon model per million mapped reads/fragment per kb of transcript per million mapped reads). The quality of the sequencing reads obtained from RNA-sequencing experiments was validated using an Excel-based differentially expressed gene analysis tool. Heatmaps were generated to visualize transcriptome differences between the control and LISSV.

### 4.5. Treatment with Recombinant PLXN4 Protein

The purified recombinant protein of human Plexin-A4 was purchased from OriGene Technologies Inc. (Rockville, MD, USA). The hUC-MSCs were cultured with a density of 1 × 10^5^ cells/35 mm dish, and the PLXNA4 recombinant protein 1.5 and 2.0 μg/100 μL medium was added 1 day later. The morphological changes in the cells were observed every day. After 5 days, cell images were obtained using an optical microscope.

### 4.6. Small Interfering RNA (siRNA) Transfection

For gene silencing, the siTran 2.0 transfection reagent from OriGene Technologies Inc. (Rockville, MD, USA) was used. In total, 8 × 10^3^ cells were seeded before transfection in 6-well culture plates, and 1 mL of growth media, 90 nmol *siPLXNA4* Oligo duplex, 2.4 μL of the siTran 2.0 transfection reagent, and 100 μL of the transfection buffer were added to the cells. To lower the cytotoxicity, the medium was replaced 24 h after the addition of the transfection complex, and the cells were harvested 24 h later. The sequences of the siRNAs were as follows: *siPLXNA4*, 5′-CGCAUAUGUCUACAAGAACCACUCT-3′; and *si**Negative* control, 5′- CGUUAAUCGCGUAUAAUACGCGUAT-3′.

### 4.7. Polymerase Chain Reaction (PCR) and Real-Time qPCR

Total RNA extraction from hUC-MSCs was performed using the Trizol solution (Qiagen, Valencia, CA, USA) and cDNA was obtained from total RNA using an Advantage RT-PCR kit (Clontech, Palo Alto, CA, USA). Table 1 shows the primer sequences used for real-time qPCR. The following genes were examined: Plexin A4 (*PLXNA4*), Formin 1 (*FMN1*), Amphiregulin (*AREG*), Stathmin 2 (*STMN2*), Serpin Family I Member 1 (*SERPINI1*), Microtubule-associated protein 2 (*MAP2*), Neuronal Differentiation 1 (*NEUROD1*), Glial fibrillary acidic protein (*GFAP*), Myelin basic protein (*MBP*), and Neurofilament-L (*NF-L*).

Real-time qPCR was performed according to the SimpliAmp Thermal Cycler (Applied Biosystems, Foster City, CA, USA), which enables real-time quantitative detection of PCR products based on SYBR green fluorescence due to the incorporation of SYBR green into double-stranded DNA. The results were analyzed by a comparative cycle threshold (CT) method for the quantification of relative gene expression for the housekeeping gene (*GAPDH*).

### 4.8. Western Blot Analysis

Cells were washed with phosphate-buffered saline (PBS) and scraped in radioimmunoprecipitation assay (RIPA) buffer (Abcam, Cambridge, UK) containing protease and a phosphatase inhibitor cocktail (Abcam, Cambridge, UK). Protein concentrations of total lysates were determined using the bicinchoninic acid (BCA) protein assay (Pierce Biotechnology, Rockford, IL, USA). For Western blot analysis, 1 μg/μL of proteins was loaded on 10% polyacrylamide gels, and the blots were transferred to nitrocellulose membranes. To prevent non-specific binding of the antibody to the membranes, the nitrocellulose membranes were shaken with 5% skim milk in Tris-acetate-EDTA (TAE) buffer for 30 min, followed by incubation with 1st antibodies such as anti-SERPINI1 (Santa Cruz Biotechnology Inc., Dallas, TX, USA), anti-PLXNA4 (R&D systems, Minneapolis, MN, USA), anti-FMN1 (Abcam, Cambridge, UK), anti-AREG (Abcam, Cambridge, UK), anti-FYN (Cell Signaling Technology, Boston, MA, USA), and β-actin (Sigma, St. Louis, MO, USA) at the appropriate dilutions overnight at 4 °C. The 1st antibody was diluted using the TAE buffer with 5% bovine serum albumin (Sigma-Aldrich, St. Louis, MO, USA). After three washes, membranes were incubated with anti-mouse and anti-rabbit secondary antibodies in 5% skim milk in the TAE buffer, then exposed to the SuperSignal West Femto Maximum Sensitivity Substrate (Thermo Scientific, Rockford, IL, USA) and autoradiographically imaged using the ChemiDoc XRS+ System (Bio-Rad, Hercules, CA, USA).

### 4.9. Immunofluorescence Imaging

In total, 5 × 10^3^ hUC-MSCs were cultured in 12-well plates, and the recombinant PLXNA4 protein was added 1 day later. After 5 days, cells were fixed with 4% paraformaldehyde solution for 10 min. Cells were shaken with a blocking solution (TAE buffer containing 1.5% bovine serum albumin) supplemented with 0.5% Triton X-100 for cell permeability, then incubated with the primary antibody overnight at 4 °C, including the anti-MAP2, anti-NEUROD1, anti-MBP, anti-MBP antibodies (Cell Signaling Technology, Boston, MA, USA).After washing, the cells were incubated with human secondary antibodies conjugated with the Alexa Fluor 488 conjugate and Alexa Fluor 555 conjugate (Cell Signaling Technology, Boston, MA, USA) for 1 h at room temperature. Glass coverslips were then counterstained with the Vectashield Antifade mounting medium (Vector Laboratories, Burlingame, CA, USA) and mounted onto microscope slides. Cells were examined using a Ts2R-FL microscope (Nikon, Tokyo, Japan).

### 4.10. Fluorescence-Activated Cell Sorting (FACS) Analysis

Both PE-conjugated anti-CD73 and PE-conjugated anti-CD105 antibodies were purchased from Abcam (Cambridge, UK). A total of 1 × 10^5^ cells were re-suspended in 500 µL ice-cold PBS with 3% bovine serum albumin (Sigma-Aldrich, St. Louis, MO, USA) and incubated with antibodies for overnight at 4 °C. After washing, flow cytometry was performed using the NovoCyte Flow Cytometer (Agilent Technologies, Santa Clara, CA, USA), and the data were analyzed using Novoexpress software (Agilent Technologies, Santa Clara, CA, USA).

### 4.11. Statistical Analysis

Data are shown as the mean ± SD of three independent experiments. An analysis of variance followed by Tukey’s multiple comparisons test was performed using GraphPad Prism (La Jolla, CA, USA). Mean differences are shown to be significant at * *p* < 0.05, and ** *p* < 0.01.

## Figures and Tables

**Figure 1 ijms-23-01522-f001:**
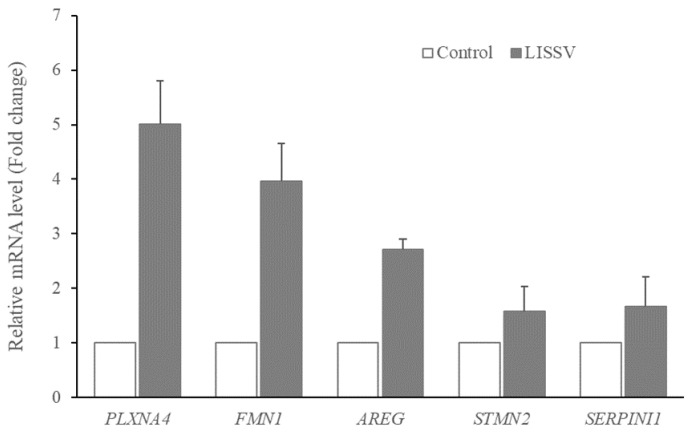
Expression of genes in human umbilical-cord-derived mesenchymal stem cells after low-intensity sub-sonic vibration treatment using RNA sequencing. Cells were harvested 4 days after LISSV treatment. Fold change in expression of Plexin-A4 (*PLXNA4*), Formin 1 (*FMN1*), Amphiregulin (*AREG*), Stathmin 2 (*STMN2*), and Serpin Family I Member 1 (*SERPINI1*). Assayed using real-time polymerase chain reaction. Column heights correspond to mean values and error bars to standard deviations (*n* = 3).

**Figure 2 ijms-23-01522-f002:**
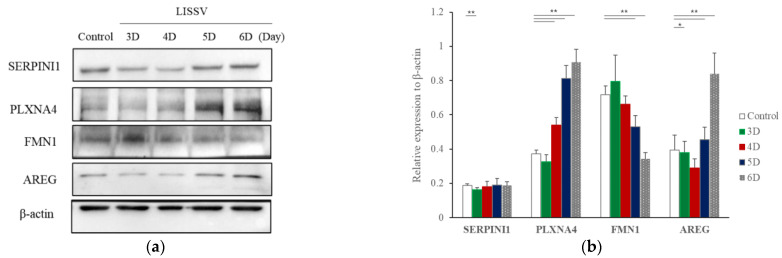
Expression of each protein in human umbilical-cord-derived mesenchymal stem cells after low-intensity sub-sonic vibration treatment. Proteins detected were Serpin Family I Member 1 (SERPINI1), Plexin A4 (PLXNA4), Formin 1 (FMN1), and Amphiregulin (AREG). (**a**) Western blot image. (**b**) Intensities of each Western blot band were quantified by Image J. Each band was normalized using β-actin. Column heights correspond to mean values and error bars to standard deviations (*n* = 3). * *p* < 0.05, ** *p* < 0.01.

**Figure 3 ijms-23-01522-f003:**
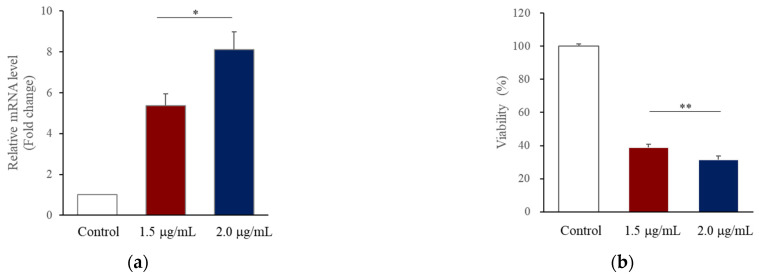
*PLXNA4* gene expression after recombinant PLXNA4 protein treatment in human umbilical-cord-derived mesenchymal stem cells. Cells were harvested 4 days after recombinant PLXNA4 protein treatment. (**a**) Real-time PCR data of each protein. (**b**) MTT data show the inhibition of proliferation. Column heights correspond to mean values and error bars to standard deviations (*n* = 3). * *p* < 0.05, ** *p* < 0.01.

**Figure 4 ijms-23-01522-f004:**
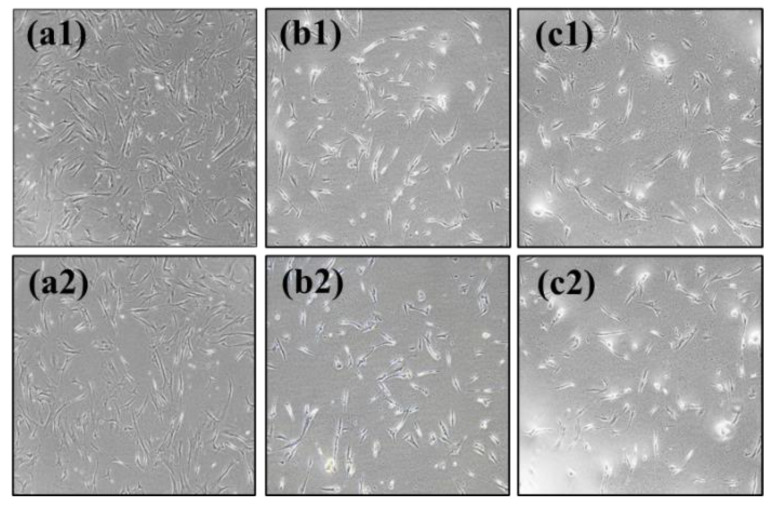
Morphological changes after recombinant PLXNA4 protein treatment for 5 days in human umbilical-cord-derived mesenchymal stem cells. (**a1**,**a2**): Untreated control cells. (**b1**,**b2**): 1.5 μg of protein-treated cells. (**c1**,**c2**): 2.0 μg of protein-treated cells. Original magnification 40×.

**Figure 5 ijms-23-01522-f005:**
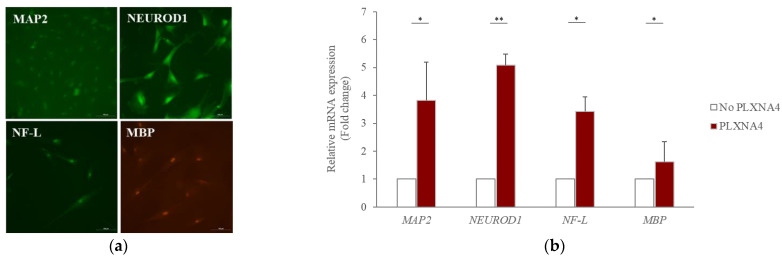
Expression of neuron-related proteins and genes in human umbilical-cord-derived mesenchymal stem cells after recombinant PLXNA4 protein treatment. Cells were harvested 5 days after 2 μg/mL recombinant PLXNA4 protein treatment. (**a**) Fluorescence images of each protein. Original magnification 200×. (**b**) Fold expression of each gene using real-time polymerase chain reaction analysis. MAP2: Microtubule-associated protein 2. NEUROD1: Neuronal Differentiation 1, NF-L: Neurofilament-L, MBP: Myelin basic protein, Column heights correspond to mean values and error bars to standard deviations (*n* = 3). * *p* < 0.05, ** *p* < 0.01.

**Figure 6 ijms-23-01522-f006:**
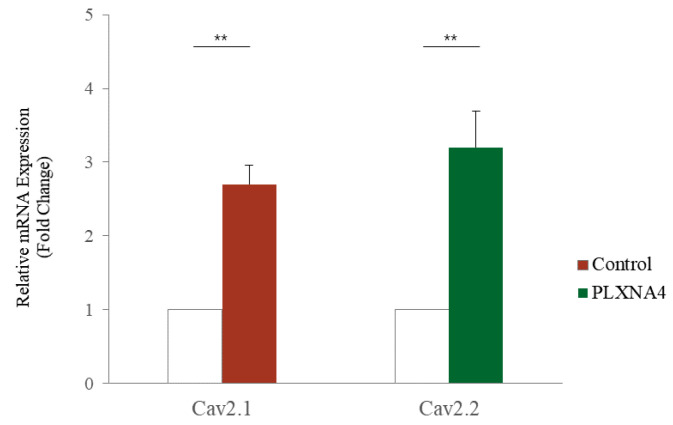
Expression of calcium channels in human umbilical-cord-derived mesenchymal stem cells after recombinant PLXNA4 protein treatment. Cells were harvested 5 days after 2 μg/mL recombinant PLXNA4 protein treatment. Fold expression of each gene using real-time polymerase chain reaction analysis. Cav2.1: Voltage-gated P/Q type calcium channel. Cav2.2: Voltage-gated N-type calcium channel, Column heights correspond to mean values and error bars to standard deviations (*n* = 3). ** *p* < 0.01.

**Figure 7 ijms-23-01522-f007:**
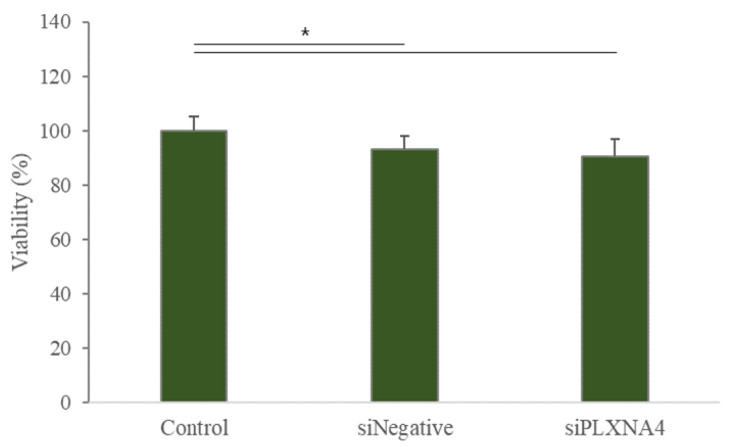
MTT data for siRNA transfection in human umbilical-cord-derived mesenchymal stem cells. Cells were harvested 2 days after siRNA duplex transfection. Column heights correspond to mean values and error bars to standard deviations (*n* = 3). * *p* < 0.05.

**Figure 8 ijms-23-01522-f008:**
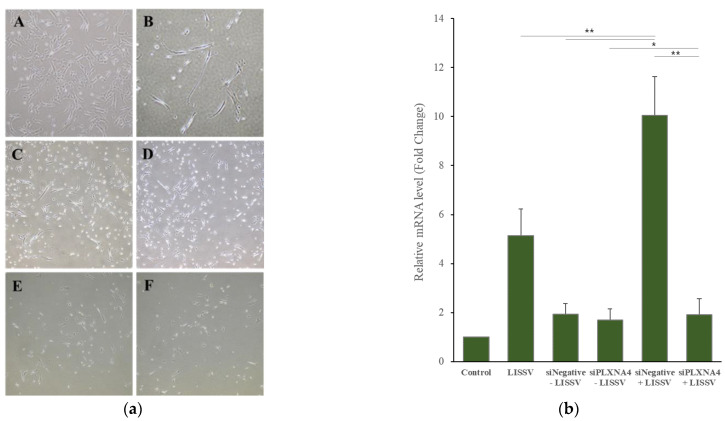
*PLXNA4* gene silencing using small interfering RNA transfection methods in human umbilical-cord-derived mesenchymal stem cells. Cells were harvested 4 days after low-intensity sub-sonic vibration treatment. (**a**) Morphology of each sample: A. Control; B. low-intensity sub-sonic vibration (LISSV); C. *siNegative* duplex 90 nmol without (−) LISSV; D. *siPLXNA4* duplex 90 nmol without (−) LISSV; E. *siNegative* duplex with (+) LISSV; F. *siPLXNA4* duplex with (+) LISSV. Original magnification 40×. (**b**) Fold change in the expression *PLXNA4* determined using real-time polymerase chain reaction analysis. Column heights correspond to mean values and error bars to standard deviations (*n* = 3). * *p* < 0.05, ** *p* < 0.01.

**Figure 9 ijms-23-01522-f009:**
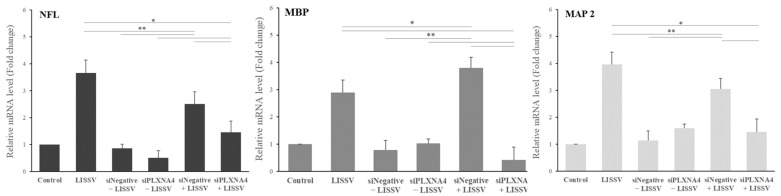
Expression of neuron-associated genes in human umbilical-cord-derived mesenchymal stem cells after siPLXNA4 duplex transfection using real-time polymerase chain reaction analysis. Cells were harvested 4 days after low-intensity sub-sonic vibration treatment. NF-L: Neurofilament-L, MBP: Myelin basic protein, MAP2: Microtubule-associated protein 2. Column heights correspond to mean values and error bars to standard deviations (*n* = 3). * *p* < 0.05, ** *p* < 0.01.

**Figure 10 ijms-23-01522-f010:**
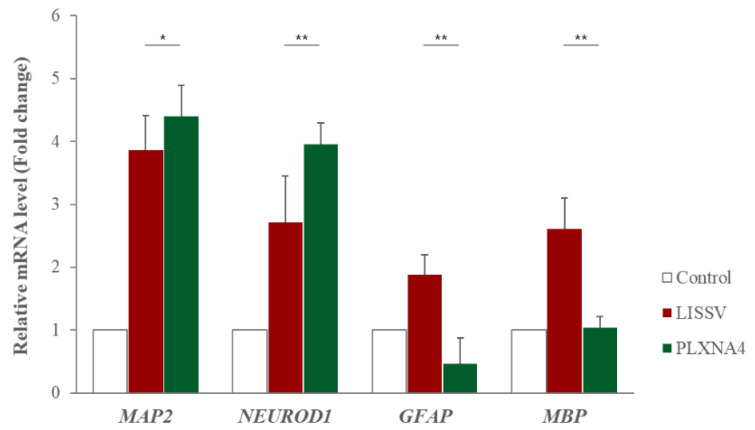
Expression of neural differentiation-specific genes in human umbilical-cord-derived mesenchymal stem cells after low-intensity sub-sonic vibration versus recombinant PLXNA4 protein. Cells were harvested 4 days after low-intensity sub-sonic vibration treatment and 5 days after recombinant PLXNA4 protein treatment. The fold change in the expression of each gene was analyzed using a real-time polymerase chain reaction. *MAP2*: Microtubule-associated protein 2. *NEUROD1*: Neuronal Differentiation 1, *GFAP*: Glial Fibrillary Acidic Protein, *MBP*: Myelin basic protein. Column heights correspond to mean values and error bars to standard deviations (*n* = 3). * *p* < 0.05, ** *p* < 0.01.

**Figure 11 ijms-23-01522-f011:**
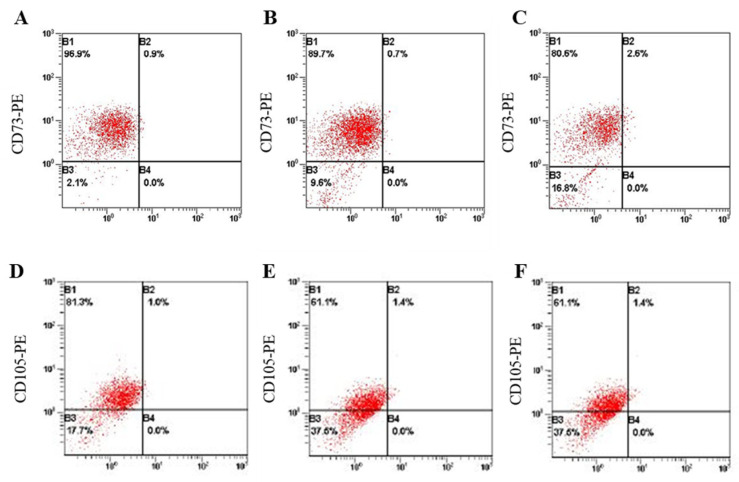
The changes of specific markers in human umbilical-cord-derived mesenchymal stem cells by low-intensity sub-sonic vibration and recombinant PLXNA4 protein treatment using FACS analysis. Cells were harvested 3 days after each treatment. PE-conjugated anti-CD73 and PE-conjugated anti-CD105 were used. (**A**,**D**) Control. (**B**,**E**) LISSV treatment. (**C**,**F**) 2 μg/mL recombinant PLXNA4 protein treatment.

**Figure 12 ijms-23-01522-f012:**
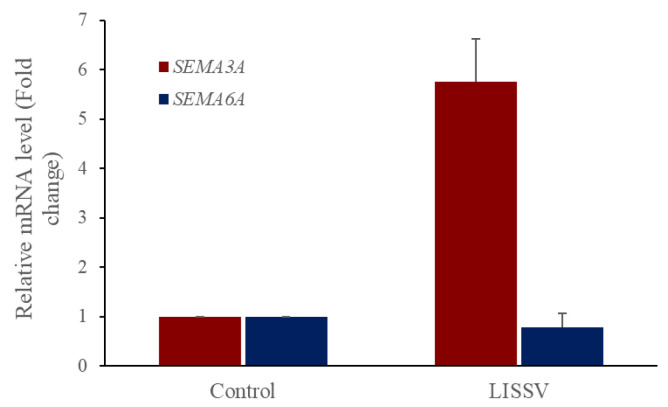
Expression of PLXNA4-dependent semaphorin signaling molecules in human umbilical-cord-derived mesenchymal stem cells by low-intensity sub-sonic vibration using real-time polymerase chain reaction analysis. Cells were harvested 4 days after low-intensity sub-sonic vibration treatment. *SEMA3A*: Semaphorin 3A, *SEMA6A*: Semaphorin 6A. Column heights correspond to mean values and error bars to standard deviations (*n* = 3).

**Figure 13 ijms-23-01522-f013:**
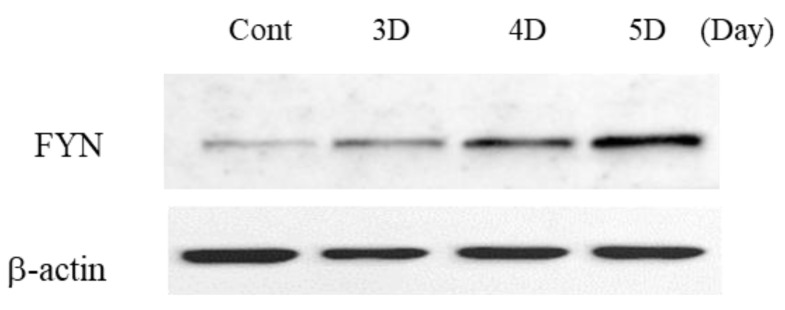
Expression of PLXNA4-dependent semaphorin signaling protein in human umbilical-cord-derived mesenchymal stem cells by low-intensity sub-sonic vibration. FYN: Src Family of nonreceptor tyrosine kinase *p59*.

**Figure 14 ijms-23-01522-f014:**
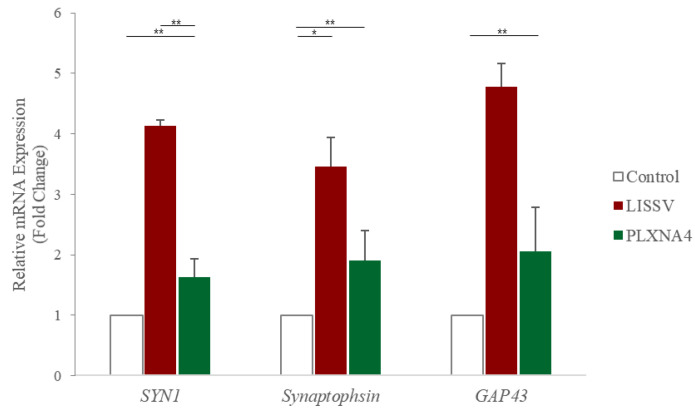
Expression of presynaptic vesicle protein-associated genes in human umbilical-cord-derived mesenchymal stem cells after low-intensity sub-sonic vibration versus recombinant PLXNA4 protein. Cells were harvested 4 days after low-intensity sub-sonic vibration treatment and 5 days after recombinant PLXNA4 protein treatment. The fold change in the expression of each gene was analyzed using real-time polymerase chain reaction. *SYN1*: Synapsin 1, GAP43: Growth-Associated Protein 43, Column heights correspond to mean values and error bars to standard deviations (*n* = 3). * *p* < 0.05, ** *p* < 0.01.

**Table 1 ijms-23-01522-t001:** Primer sequences used for RT-PCR.

**Genes**	**Upstream Primer Sequence**	**Downstream Primer Sequence**
*PLXNA4*	5′-ATC TCC GTC TCT CAG TAC AA-3′	5′-GTG ATA GGC TTG ATC ACC TC-3′
*FMN1*	5′-CCA TCA CCG TTT TCT TCT TC-3′	5′-AGT TAC AGT GCC CTT GTA TG-3′
*AREG*	5′-TTC TAG TAG TGA ACC GTC CT-3′	5′-AGA CAT AAA GGC AGC TAT GG-3′
*STMN2*	5′-CAG AGG GAA GGA GAG AAG CAA T-3′	5′-TCA TTA GGC AAT GGT GGG TT-3′
*SERPINI1*	5′-AAA ACC TCT CGG GTG AAA G-3′	5′-GCT GTC ATA TCC CAT TGA GT-3′
*MAP2*	5′-CTC AAC AGT TCT ATC TCT TCT TCA-3′	5′-TCT TCT TGT TTA AAA TCC TAA CCT-3′
*NFL*	5′-CAA GAA CAT GCA GAA CGC TG-3′	5′-GCC TTC CAA GAG TTT CCT GT-3′
*NEUROD1*	5′-ACA GTC ACC AGT GTG GTG GA-3′	5′-CGT AGC CTC TGG AGA ACC TG-3′
*GFAP*	5′-TCATCGCTCAGGAGGTCCTT-3′	5′-CTGTTGCCAGAGATGGAGGTT-3′
*MBP*	5′-CGG CAA CTA CGT GCT CTT CA-3′	5′-GTG ACT TCA TCT CGT GGG CC-3′
*SEMA3A*	5′-TAA GGA GAA AGG AGG AGA GGT G-3′	5′-GTG CTG GTT TGA ACT AGA GG-3′
*SEMA6A*	5′-TTA CAA CAC AGT GTA TGG GC-3′	5′-CTT TGA GGT AAC TTT CCC GA-3′
*SYN1*	5′-GCT CAA CAA ATC CCA GTC TC-3′	5′-GAG GAG TCA GGT TTC TCA AG-3′
*Synaptophysin*	5′-CCT ATA CCC TAG GTC TCC AC-3′	5′-CCT GTC CTC CTT TTA GAT CC-3′
*GAP43*	5′-CCA TGC TGT GCT GTA TGA GAA G-3′	5′-TAA GGA CTA GGT CGA ACT GC-3′
*Cav2.1*	5′-CCG TGT GAT AAG AAC TCT GG-3′	5′-GAC ATG TGT CTC AGC ATC-3′
*Cav2.2*	5′-CCA TCT TCT ACG TGG TCT AC-3′	5′-CAT CAG CTC GTA CTC ATA GG-3′
*GAPDH*	5′-ACC ACA GTC CAT GCC ATC AC-3′	5′-TCC ACC ACC CTG TTG CTG TA-3′

## Data Availability

Data available upon reasonable request.

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
