# Peer review of "Induction of PLXNA4 Gene during Neural Differentiation in Human Umbilical-Cord-Derived Mesenchymal Stem Cells by Low-Intensity Sub-Sonic Vibration"

_ijms, 2022, doi:10.3390/ijms23031522_

Round 1
Reviewer 1 Report
In this paper, the authors investigated effects of PLXNA4 gene and protein on low intensity sub-sonic vibration (LISSV)-induced neural differentiation in hUC-MSCs. This study has serious disadvantage. The results don’t support their conclusions. My comments are as follows:
- The authors need to use electrophysiological methods to demonstrate the neuronal differentiation. The images of morphology and immunostaining did not show typical neuronal cells. The increased expression level of neuron-related genes does not mean the cells already differentiated into neuronal cells. Without successful neuronal differentiation, the base of this study does exist.
- In the introduction part, the authors should describe more about the effects of low intensity sub-sonic vibration on cellular function or tissue function.
- There are many grammatical errors in the manuscript. The authors should ask an English service or a native speaker to proofread the article.
Author Response
Thanks for the good comments. You have given me the opportunity to revise this manuscript.
I will do my best to revise the manuscript.
To identify the inserted parts in this manuscript, I used blue font. The format of the figure in the original version has been changed to incorporate your comments on this manuscript into the paper.
Thank you for reading my revised manuscript.
Kind Regards,
Dr. Hyunjin Cho
_____________________________________________________________________________________
- The authors need to use electrophysiological methods to demonstrate the neuronal differentiation. The images of morphology and immunostaining did not show typical neuronal cells. The increased expression level of neuron-related genes does not mean the cells already differentiated into neuronal cells. Without successful neuronal differentiation, the base of this study does exist.
; In a previous report, we reported the differentiation of hUC-MSCs by low-intensity subsonic oscillations (Cho et.al., Life Sciences 2012, 90, 591-599). The differentiation by LISSV was induced in LDMEM medium without any supplements for neural induction. That was non-specific, so genes related to oligodendrocytes, astrocytes and neurons were all expressed.
In this report, one of the genes related to axon induction, the PLXNA4 gene, was induced by LISSV. In this experiment, any neural differentiation in hUC-MSCs by LISSV was not observed in the experiment in which the PLXNA4 gene was knocked down.
For specific neuronal induction of hUC-MSCs by LISSV, it is thought possible by mixing PLXNA4 and an appropriate supplement in the medium. We are currently conducting experiments using various types of supplements to induce differentiation of specific lineages of neurons. This particular neuronal lineage may be a functional neuron, requiring electrophysiological data to identify it. Experiments are currently underway to induce differentiation of hUC-MSCS into neurons using various mechanical stimuli and previously reported genes through various channel-related gene analyses and patch clamp method.
- In the introduction part, the authors should describe more about the effects of low intensity sub-sonic vibration on cellular function or tissue function.
; I have added these sentences in Introduction section as follows.
“Sub-sonic means a frequency that is so low that it is inaudible and slower than the speed of sound. Low-intensity vibration (LIV) is a stimulus that transmits vibration with its intensity in the range of 20 to 200 Hz frequencies. LIV can be effective in im-proving bone and muscle index at the tissue level [8-10], in reducing myeloma cell-induced osteoclast formation [11]. Also, LIV decreased the pro-inflammatory cytokines IL-6, IFN-g, and TNF-a in cultured murine macrophages [12], and inhibited tumor progression [13].”
- There are many grammatical errors in the manuscript. The authors should ask an English service or a native speaker to proofread the article.
; As you said, this article was corrected through the English service of a native speaker.
_____________________________________________________________________________________
Reviewer 2 Report
- How was the identity of the UC-MSCs confirmed? For the rest of the studies, did the authors ever re-assess the level of MSC marker expression?
- Lines 67-69 – When the authors say 16 genes were increased more than 3 fold, why are they reporting only the specific 5 genes in Fig 1 & 2? Why not list all 16?
- Why did the authors not report for Stathmin 2 for protein levels?
- The authors should combine the following figures for better visual detail:
- Fig 1 & 2 – RNA seq results
- Fig 3-5 – recombinant PLXN4 results
- Fig 6-7 – siRNA silencing results
- Fig 8 -10 – changes with PLXN4 Vs LISSV
- How many replicates are each of the results from? Please mention in figure legends.
- Recombinant PLXN4 treatment studies – How were the tested doses decided on? Did the authors test any doses outside of those presented?
- The authors need to add in additional details such as time point of harvest for all the figures.
- Fig 5 – which dose of and after how many days of PLXN4 treatment are these images from?
- Can the authors provide higher mag insets for Figs 4, 5A and 6A-F?
- Given a rather significant role for PLXN4/SEMA3 in axon guidance in the nervous system, did the authors look at any other neurite outgrowth markers such as GAP43 for e.g.?
- The authors need to present viability data from the siRNA experiments. This is critical for accurate assessment
- Line 298 – The title should be re-labelled ‘Treatment with recombinant PLXN4 protein’
Author Response
Thanks for the good comments. You have given me the opportunity to revise this manuscript.
I will do my best to revise the manuscript.
To identify the inserted parts in this manuscript, I used blue font. The format of the figure in the original version has been changed to incorporate your comments on this manuscript into the paper.
Thank you for reading my revised manuscript.
Kind Regards,
Dr. Hyunjin Cho
____________________________________________________________________________________
- How was the identity of the UC-MSCs confirmed? For the rest of the studies, did the authors ever re-assess the level of MSC marker expression?
; After treatment with LISSV and recombinant PLXNA protein, the change in the expression level of hUC-MSC markers was confirmed through FACS analysis.
Figure 10 has been inserted and the following explanation has been added in section 2.4.
Figure 10. The changes of specific markers in human umbilical cord-derived mesenchymal stem cells by low-intensity subsonic vibration and recombinant PLXNA4 protein treatment using FACS analysis. Cells were harvested 3 days after each treatment. PE-conjugated anti-CD73 and PE-conjugated anti-CD105 were used. (A, D) Control. (B, E) LISSV treatment. (C, F) 2 mg/ml recombinant PLXNA4 protein treatment.
We thought that MSC-specific markers were no longer expressed when differentiation was induced by both LISSV and recombinant PLXNA4 protein. CD73, CD105, and CD90 are markers specifically expressed in human MSCs [28]. In Fluorescence-activated cell sorting (FACS) analysis, we analyzed anti-CD73 and anti-CD105 expression after LISSV and recombinant PLXNA4 protein treatment (Fig. 11). Before treatment, anti-CD73 was expressed 97.8% in hUC-MSCs and anti-CD105 was 81.3%. Before treatment, anti-CD73 was expressed at 97.8% and anti-CD105 was expressed at 81.3% in hUC-MSCs. When LISSV was given for 3 days, the expression of anti-CD73 was reduced to 90.4%, and anti-CD105 was reduced to 62.5%. In the case of PLXNA4 treatment, the expression of anti-CD73 was reduced to 83.2%, like LISSV, and anti-CD105 was also reduced to 62.5%. Both LISSV and recombinant PLXNA4 protein treatment reduced the expression of MSC-specific markers, suggesting that the differentiation process of hUC-MSCs is in progress.
And I added these sentences to the discussion section as shown below.
In our previous report, when LISSV was given to hUC-MSCs, differentiation of hUC-MSCs was a neural non-specific process, so all three types of neural cell markers, astrocytes, oligodendrocytes, and neuron, were induced. However, PLXNA4-induced differentiation in hUC-MSCs is a neuron-specific process, as GFAP and MBP were not expressed.
hUC-MSCs are cells capable of self-renewal and differentiation into various lineages [35]. Wharton's jelly derived from the human umbilical cord contains a higher amount of primitive MSCs compared to MSCs derived from bone marrow [36]. When MSCs begin to differentiate into cells of other lineages, those cells cannot maintain their stemness. Human MSCs express CD73, CD90 and CD105, but not CD34, CD45, or CD14 [37,38]. During both LISSV and recombinant PLXNA4 protein treatment, the expression of hUC-MSC specific markers, CD73 and CD105 were reduced. Therefore, those cells have begun to differentiate into other cells, particularly nerve-like cells.
- Lines 67-69 – When the authors say 16 genes were increased more than 3 folds, why are they reporting only the specific 5 genes in Fig 1 & 2? Why not list all 16?
; In RNA sequencing analysis, 16 genes were significantly increased more than 3-fold. To confirm the increase of 16 genes, real-time PCR assay was performed on 16 genes. This assay was repeated three or more times, but only five genes were found to have increased expression. Therefore, we only mentioned five genes that were identified.
- Why did the authors not report for Stathmin 2 for protein levels?
; Stathmin 2 is a gene identified through qPCR assay. Then, the expression of protein level was confirmed using STMN2 purchased from Abcam. However, in Western blot analysis, no significant band could be found in more than 3 replicates. So, except for STMN2, the expression of the remaining four proteins was shown in Figure 2.
I hope you understand.
- The authors should combine the following figures for better visual detail:
Fig 1 & 2 – RNA seq results
Fig 3-5 – recombinant PLXN4 results
Fig 6-7 – siRNA silencing results
Fig 8 -10 – changes with PLXN4 Vs LISSV
; That's a good comment. The results were reorganized as follows.
Fig 1-2 – 2.1. RNA sequencing confirmed that six genes were increased by LISSV.
Fig 3-5 – 2.2. Recombinant PLXNA4 protein affected neural differentiation of hUC-MSCs
Fig 6-7 – 2.3. The neural differentiation of hUC-MSCs by LISSV was not induced upon PLXNA4 gene silencing
Fig 8-11 – 2.4. Changes with LISSV vs PLXNA4 in sema-dependent signaling
- How many replicates are each of the results from? Please mention in figure legends.
; I have added the sentence "Column heights correspond to mean values ​​and error bars to standard deviations (n=3)" in Figure 1,2,3,5,6,7,8,9 legend.
- Recombinant PLXN4 treatment studies – How were the tested doses decided on? Did the authors test any doses outside of those presented?
; To confirm the cytotoxicity of recombinant PLXNA4 protein in UC-MSC, MTT analysis was performed from 0.5 mg/ml to m3 g/ml. hUC-MSC was not toxic up to 2mg/ml, and the number of viable cells was significantly reduced to 77% at 3mg/ml.
In section 2.2, I have added, " The recombinant PLXNA4 protein was non-toxic in hUC-MSCs up to 2 mg/ml and significantly reduced the number of viable cells to 77% at 3 mg/ml (data was not shown). "
- The authors need to add in additional details such as time point of harvest for all the figures.
; I have added the cell harvest time points to the legend of Figures 1,3,5,6,7,8,9.
- Fig 5 – which dose of and after how many days of PLXN4 treatment are these images from?
; 2 mg of recombinant PLXNA4 protein was treated in Fig 5A.
I have added this sentence in Fig 5A legend
“Figure 5. Expression of neuron-related proteins and genes in human umbilical cord-derived mesenchymal stem cells after recombinant PLXNA4 protein treatment. Cells were harvested 5 days after 2 mg/ml recombinant PLXNA4 protein treatment. (a) Fluorescence images of each protein. Original magnification 200×. (b) Fold expression of each genes using real-time polymerase chain reaction analysis. MAP2: Microtubule-associated protein 2. NEUROD1: Neuronal Differentiation 1, NF-L: Neurofilament-L, MBP: Myelin basic protein, Column heights correspond to mean values and error bars to standard deviations (n=3). *p < 0.05, **p < 0.01.”
- Can the authors provide higher mag insets for Figs 4, 5A and 6A-F?
; Sorry, I do not have a higher magnification of those images, so I could not insert them.
I hope you understand.
- Given a rather significant role for PLXN4/SEMA3 in axon guidance in the nervous system, did the authors look at any other neurite outgrowth markers such as GAP43 for e.g.?
; After treatment with LISSV and recombinant PLXNA protein, the change in the expression level of presynaptic vesicle related genes for neurite outgrowth were confirmed using qPCR.
Figure 13 has been inserted and the following explanation has been added in section 2.4.
Figure 13. Expression of presynaptic vesicle protein associated genes in human umbilical cord-derived mesenchymal stem cells after low-intensity subsonic vibration versus recombinant PLXNA4 protein. Cells were harvested 4 days after low-intensity subsonic vibration treatment and 5 days after recombinant PLXNA4 protein treatment. The fold change in expression of each gene was analyzed using real-time polymerase chain reaction. SYN1: Synapsin 1, GAP43: Growth Associated Protein 43, Column heights correspond to mean values and error bars to standard deviations (n=3). *p < 0.05, **p < 0.01.
Presynaptic vesicles for neurite outgrowth in neurons are induced through activation of this signal. GAP43 regulates presynaptic vesicle interactions and SYN1 and synaptophysin proteins are presynaptic vesicle proteins. LISSV and recombinant PLXNA4 protein increased the expression levels of SYN1, GAP43, and synaptophysin genes (Fig.13). The expression of neuron-related proteins was induced by activation of the semaphorin 3A-dependent plexin-A4 signaling cascade by LISSV in hUC-MSCs.
And I added these sentences to the discussion section as shown below.
Sema3A regulates the density of the dendritic spine, small membrane protrusions from dendrites of neurons [46,47], and this signaling activation induces growth cone collapse, induce neuronal cells [48,49]. SYN1 and Synaptophysin are presynaptic vesicle proteins located in the cytoplasmic membrane of the presynapse, and GAP43, a neuromodulin, plays a role in regulating presynaptic vesicular function and axonal growth [50,51]. Expression of SYN1, GAP43, and synaptophysin was induced in hUC-MSCs through sema3 signaling activation upon treatment with LISSV and re-combinant PLXNA4 protein.
- The authors need to present viability data from the siRNA experiments. This is critical for accurate assessment
: As you mentioned, we tested the cytotoxicity of siRNA duplex transfection in hUC-MSCs
Figure 6 has been inserted and the following explanation has been added in section 2.3.
Figure 6. MTT data for siRNA transfection in human umbilical cord-derived mesenchymal stem cells. Cells were harvested 2 days after siRNA duplex transfection. Column heights correspond to mean values and error bars to standard deviations (n=3). *p < 0.05.
We tested the cytotoxicity of siRNA duplex transfection in hUC-MSCs before siRNA experiments. When treated with 90 nmol of siPLXNA4 duplex transfection, viable cells were 93.1% after siNegative control transfection and 90.4% after siPLXNA4 transfection (Fig. 6).
- Line 298 – The title should be re-labelled ‘Treatment with recombinant PLXN4 protein’
; I have corrected line 298 “4.5 Treatment with recombinant PLXN4 protein”
_____________________________________________________________________________________
Round 2
Reviewer 1 Report
The authors answered my questions and improved their manuscript. However, some important places are still not improved.
- Figure 4 and 5 still did not show morphological and immunological changes of neuronal cells. They just look like normal fibroblasts. The authors should finish their electrophysiological experiments and provide the data.
- Results 2.2.: Since cell viability of PLXNA4-treated cells was markedly decreased at concentrations of 1.5 and 2.0 ug/ml after 4 days of culture, how can the authors say “The recombinant PLXNA4 protein was non-toxic in hUC-MSCs up to 2 ug/ml…”?
- The concentrations of proteins or other reagents should be expressed as ug/ml instead of ug.
Author Response
- Figure 4 and 5 still did not show morphological and immunological changes of neuronal cells. They just look like normal fibroblasts. The authors should finish their electrophysiological experiments and provide the data.
; A qPCR method was performed to analyze the expression of voltage-gated calcium channel-related genes via electrophysiological experiments. Cav2.1 and Cav2.2 are voltage-gated calcium channels, which are key mediators that induce changes in membrane potential. After 5 days of treatment with recombinant PLXNA4 protein, cells were harvested and qPCR was performed. The expression of these two genes was increased. We've inserted these results and comments into the Results and Discussion section.
Results:
Voltage-gated calcium (Ca2+) channels are key channels that induce changes in membrane potential, and intracellular Ca2+ transients signal the initiation of many physiological events. The Cav2 subfamily is primarily responsible for the initiation of synaptic transmission [28]. Recombinant PLXNA4 protein has induced the expression of Cav2.1 and Cav2.2 genes in hUC-MSCs (Fig. 6).
Figure 6. Expression of calcium channels in human umbilical cord-derived mesenchymal stem cells after recombinant PLXNA4 protein treatment. Cells were harvested 5 days after 2 µg/ml recombinant PLXNA4 protein treatment. Fold expression of each gene using real-time polymerase chain reaction analysis. Cav2.1: Voltage-gated P/Q type calcium channel. Cav2.2: Voltage-gated N-type calcium channel, Column heights correspond to mean values and error bars to standard deviations (n=3). **p < 0.01.
Discussion:
Voltage-gated calcium channels are the main mediators that allow calcium to flow into neurons when depolarization occurs. The Cav2.1 is P/Q-type calcium channel, and Cav2.2 is N-type calcium channel, P/Q and N channels trigger neurotransmitter release [36]. Recombinant PLXNA4 protein induced neural differentiation while expressing Cav2.1 and Cav2.2.
- Results 2.2.: Since cell viability of PLXNA4-treated cells was markedly decreased at concentrations of 1.5 and 2.0 ug/ml after 4 days of culture, how can the authors say “The recombinant PLXNA4 protein was non-toxic in hUC-MSCs up to 2 ug/ml…”?
; In the first revision, reviewers noted cytotoxicity data of recombinant PLXNA4 protein in hUC-MSCs. Cytotoxicity data of recombinant proteins added to cells are very important.
In general, results of cytotoxicity by added proteins or other reagents can be obtained from MTT results after 2~3 days of treatment. If the added protein is cytotoxic, the number of cells will be reduced.
So I have inserted “The recombinant PLXNA4 protein was non-toxic in hUC-MSCs up to 2 mg/ml and significantly reduced the number of viable cells to 77% at 3 mg/ml (data was not shown).”
But I already mentioned in this article “The cells did not proliferate when treated with recombinant PLXNA4 protein compared to controls, as they did with LISSV exposure.”
After 5 days of MTT treatment, the non-proliferation of cells means that the number of cells did not increase compared to the control cells, and no dead cells were observed at this time. Therefore, I noted that "cell proliferation was inhibited without dead cells."
If I am lacking in explanation, I will correct it.
- The concentrations of proteins or other reagents should be expressed as ug/ml instead of ug.
; I have corrected Figure 3.
Figure 3. PLXNA4 gene expression after recombinant PLXNA4 protein treatment in human umbilical cord-derived mesenchymal stem cells. Cells were harvested 4 days after recombinant PLXNA4 protein treatment. (a) Real-time PCR data of each protein. (b) MTT data shows the inhibition of proliferation. Column heights correspond to mean values and error bars to standard deviations (n=3). **p < 0.01.
_________________________________________________________________________________________________
Reviewer 2 Report
I'm not sure how the authors may have addressed the point about combining figures for better visual detail. The version I have received do not seem to have the combined version of the figures as the authors claimed in response #4.
Author Response
Thanks for the good comments.
I will do my best to revise the manuscript. If I am lacking in explanation, I will correct it.
Thank you for reading my revised manuscript.
Kind Regards,
Dr. Hyunjin Cho
_____________________________________________________________________________________
I'm not sure how the authors may have addressed the point about combining figures for better visual detail. The version I have received do not seem to have the combined version of the figures as the authors claimed in response #4.
; In this 2nd revision, reviewers noted the electrophysiology data of recombinant PLXNA4 protein treatment in hUC-MSCs. So I added the data and reorganize it like this.
Fig 1-2 – 2.1. RNA sequencing confirmed that six genes were increased by LISSV.
Fig 3-6 – 2.2. Recombinant PLXNA4 protein affected neural differentiation of hUC-MSCs
Fig 7-9 – 2.3. The neural differentiation of hUC-MSCs by LISSV was not induced upon PLXNA4 gene silencing
Fig 10-14 – 2.4. Changes with LISSV vs PLXNA4 in sema-dependent signaling
_____________________________________________________________________________________
Round 3
Reviewer 1 Report
The authors answered my questions and improved their manuscript. The manuscript can be accepted for publication.